**Data Availability Statement:** All relevant data are within the paper and its Supporting information files.

**Funding:** This work was supported by grants from the National Natural Science Foundation of China

# DPP-4 inhibitors may improve the mortality of coronavirus disease 2019: A meta-analysis

**Yan Yang☯, Zixin Cai☯, Jingjing Zhang🄳 ***

National Clinical Research Center for Metabolic Diseases, Metabolic Syndrome Research Center, Key Laboratory of Diabetes Immunology, Ministry of Education, and Department of Metabolism and Endocrinology, The Second Xiangya Hospital of Central South University, Changsha, Hunan, China

☯ These authors contributed equally to this work.

* Doctorzhangjj@csu.edu.cn

## Abstract

### Aims

DPP-4 inhibitors are predicted to exert a protective effect on the progression of coronavirus disease 2019 (COVID-19). We conducted this meta-analysis to investigate this hypothesis.

### Methods

Four databases, namely, PubMed, Web of Science, EMBASE and the Cochrane Library, were used to identify studies on DPP-4 and COVID-19. The outcome indicators were the mortality of COVID-19. Funnel plots, Begg's tests and Egger's tests were used to assess publication bias.

### Results

Four articles were included with a total of 1933 patients with COVID-19 and type 2 diabetes. The use of DPP-4 inhibitors was negatively associated with the risk of mortality (odds ratio (OR) = 0.58 95% confidence interval (CI), 0.34–0.99).

### Conclusions

DPP-4 inhibitors may improve the mortality of patients with COVID-19 and type 2 diabetes. As few relevant studies are available, more large-scale studies need to be performed.

## Introduction

A global pandemic of coronavirus disease 2019 (COVID-19) began in 2020. COVID-19 is caused by severe acute respiratory syndrome coronavirus 2 (SARS-CoV-2) [1]. The wide-spread COVID-19 pandemic is reminiscent of two past epidemics of respiratory diseases caused by coronaviruses, the severe acute respiratory syndrome (SARS) epidemic in 2002 [2] and the Middle East respiratory syndrome (MERS) epidemic in 2012 [3]. The three major infectious respiratory diseases caused by coronaviruses that have caused epidemics in the 21st

[82070807, 91749118, 81770775, and 81730022], the Planned Science and Technology Project of Hunan Province [2017RS3015] and National Key Research and Development Program [2019YFA0801903 and 2018YFC2000100].

**Competing interests:** The authors have declared that no competing interests exist.

century are SARS, MERS and COVID-19. Because SARS-CoV and MERS-CoV enter and infect cells via dipeptidyl peptidase-4 (DPP-4) [4, 5], SARS-CoV-2 may also enter cells by binding to DPP-4. However, recent studies have shown that the SARS-CoV-2 spike protein does not interact with human membrane-bound DPP-4 (CD26) [6, 7]. Although DPP-4 does not function as the receptor in SARS-CoV-2 infections, DPP-4 inhibitors (DPP-4is), one of the new oral therapies for diabetes characterized by neutral weight and few adverse effects, is now used to improve insulin secretion as a treatment for T2DM [8], and researchers have speculated on whether DPP-4 inhibitors (DPP-4i) play a role in protecting against COVID-19 and their use as therapeutic drugs to improve outcomes in patients with COVID-19 and type 2 diabetes (T2DM) [9, 10].

An increasing number of studies have shown that T2DM is the comorbidity with the strongest negative effect on the prognosis of patients with COVID-19. Patients with T2DM who contract COVID-19 have a higher mortality rate and are more likely to develop severe COVID-19 [11, 12]. The collision of these two major global epidemics suggests that the correct use of anti-diabetic agents is an urgent issue that must be addressed. As DPP-4is are commonly used hypoglycemic agents, the relationship between DPP-4i use and COVID-19 has also attracted increasing attention, we conducted this meta-analysis to determine whether DPP-4is exert a protective effect on the development of COVID-19 mortality.

Although recent observational studies have described the relationship between the use of DPP-4is and COVID-19 [13, 14], no meta-analysis has been performed to synthesize this evidence. The purpose of this article was to systematically describe the relationship between the use of DPP-4is and the mortality of COVID-19 and provide evidence that can be used to guide the treatment of patients with diabetes during the COVID-19 pandemic.

## Methods

This meta-analysis was conducted according to the Preferred Reporting Items for Systematic Reviews and Meta-Analyses (PRISMA) statement guidelines, as described previously [15].

### Article search strategy

We searched for articles published between September 28, 2020, and October 30, 2020. The PubMed (2013–2020, October 30), Cochrane Library (1960–2020, October 30), EMBASE (1960–2020, October 30) and Web of Science (1950–2020, October 30) databases were searched in this study. Searches for all published articles related to both DPP-4 and COVID-19 were performed. The following search terms were used: "dipeptidyl peptidase-4 inhibitors", "Dpp4", "DPP-4", "saxagliptin", "alogliptin", "sitagliptin", "linagliptin", "vildagliptin", "SARS", "COVID-19", "SARS-CoV-2", and ''2019 novel coronavirus". Additional papers were identified by performing manual searches of the reference lists of relevant articles and tracking citations.

### Selection criteria

Two reviewers (YY and ZC) independently reviewed all the eligible studies and selected those suitable for inclusion. Disagreements were settled by reaching a consensus or with the help of a third reviewer (JZ). All the articles included in this meta-analysis met the following criteria: (1) they contained information on DPP-4is and the outcomes of COVID-19, including mortality and the development of severe COVID-19; and (2) the subjects were patients with both COVID-19 and T2DM. Articles were excluded if they met the following criteria: (1) they lacked information or data necessary for the purpose of this meta-analysis and (2) they were published as letters, reviews, editorials, or conference abstracts.

### Data extraction

All relevant articles were imported into EndNote X9 software and reviewed independently by two authors (YY and ZC). Discrepancies between authors were settled with the help of a third reviewer (JZ). The following information was extracted from the selected studies by two independent investigators: author, year, country, type of study, age, sample size, population and COVID-19 outcomes. All the extracted data were then imported into Excel.

### Quality assessment of included studies

The quality of the included studies was assessed using the Newcastle-Ottawa Scale (NOS) [16]. We assessed the quality of all relevant studies based on the type of study, sample size, participant selection, representativeness of the sample, adequacy of follow-up, comparability (exposed-unexposed or case-control), and method of ascertaining cases and controls. A study with a score of 6 or more was defined as a high-quality study. The possible range of NOS scores is 0 to 9; studies scoring at least 7 have the lowest risk of bias. Those that scored 4–6 are assigned a modest risk of bias, and those scored <3 have the highest risk of bias.

### Statistical analysis

All analyses were performed using Stata software (version 13.0). The correlations between DPP-4is and adverse outcomes were reported as the pooled odds ratios (ORs) and 95% confidence intervals (CIs). ORs > 1 represented a direct association, and those < 1 represented an inverse association. All results of the included studies were analyzed with random-effects models. $I^2$ statistics were used to assess the degree of heterogeneity: 25%, 50%, and 75% represented low, moderate, and high degrees of heterogeneity, respectively. Begg's and Egger's tests and funnel plots were used to detect potential publication bias, with a p-value <0.05 suggesting the presence of bias. The trim-and-fill method was also used to obtain an adjusted effect size when publication bias was detected.

## Results

### Search results and study characteristics

The flowchart of the study selection process is shown in Fig 1. After a preliminary search of the selected electronic databases, 475 studies were identified. Then, 119 duplicates were eliminated. After further excluding 329 studies based on their titles and abstracts, 27 articles remained. Of those 27 articles, 23 were excluded after the full text was read for the following reasons: (1) insufficient participant information was provided (n = 12); (2) the original data regarding DPP-4i use were not provided (n = 7); and (3) the outcome of COVID-19 was the severity instead of the mortality of the disease (n = 4). Finally, 4 articles related to the use of DPP-4is and COVID-19 were included in this meta-analysis. The basic characteristics of the studies are shown in Table 1. Among the 4 studies included in this analysis, 1 was performed in France and 3 were performed in Italy (Table 1). All 4 included studies were published in 2020.

### Quality assessment

The NOS mainly consists of three parts: sample selection, comparability of cases and controls, and exposure. All four included studies had NOS scores higher than 8, indicating no risk of bias in our analysis. Details of the risk of bias assessment are described in Table 2.

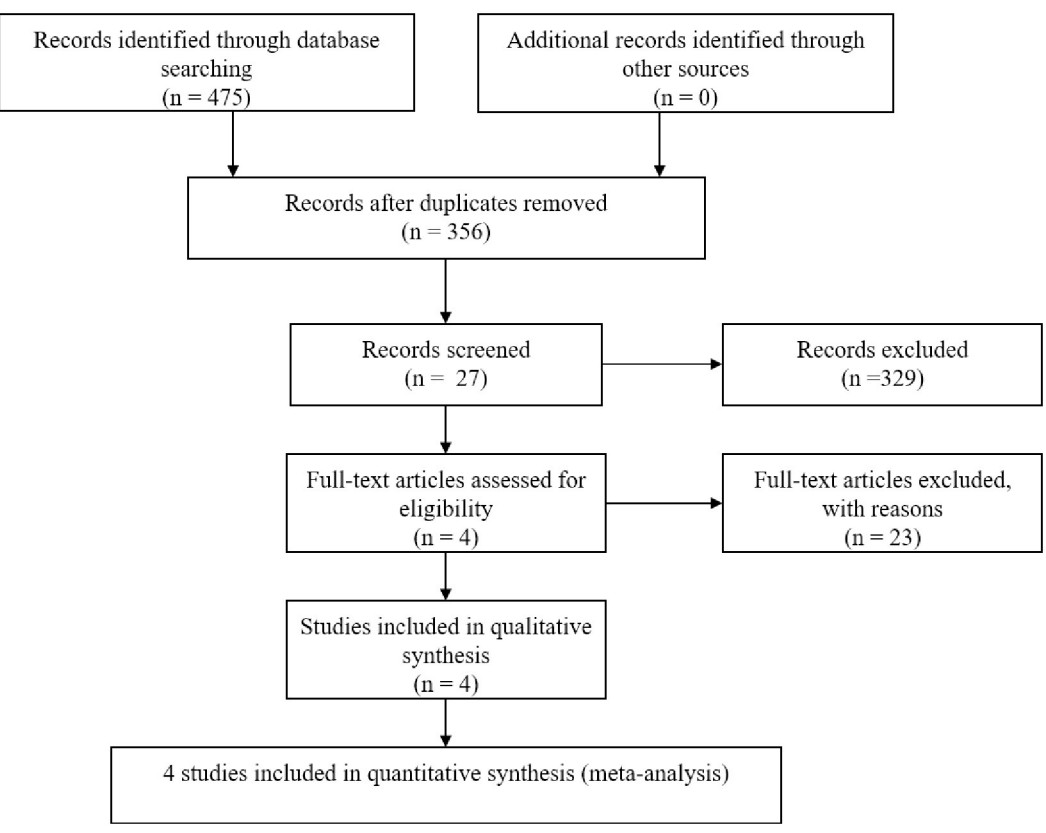

**Fig 1. Flow diagram of the study selection process.**

## DPP-4i use and COVID-19 mortality

The results of the meta-analysis of the use of DPP-4is and the mortality of COVID-19 are shown in Fig 2. In general, the use of DPP-4is was associated with decreased mortality due to COVID-19 (OR = 0.58 95% CI, 0.34–0.99); No significant heterogeneity was observed ($I^2$ statistic = 51.1%, p = 0.105) (Fig 2). The results of Egger's and Begg's tests (p>0.05) and an inspection of the funnel plots showed that publication bias did not exist among the studies (Fig 3). A sensitivity analysis was conducted by omitting one study at a time and showed that the results were stable (Fig 4).

**Table 1. Description of eligible studies reporting the association between DPP-4i and the mortality of COVID-19.**

| Author | Year | Country | Age | Type of study | Kind of DPP4i | Time of DPP4i treatment | Sample size | Control group | Population |
|---|---|---|---|---|---|---|---|---|---|
| Cariou [14] | 2020 | French | 69.8 ± 13.0 | A nationwide multicentre observational study | ND | prior to hospitalization | 1317 | ND | Diabetes with COVID-19 |
| Fadini [13] | 2020 | Italy | 70.3 | A case-control study | ND | prior to hospitalization | 85 | No DPP4i use | T2D with COVID-19 |
| Solerte [36] | 2020 | Italy | 69 | A Multicenter, Case-Control, Retrospective, Observational Study | sitagliptin | at the Time of Hospitalization | 338 | Insulin treatment | T2D with COVID-19 |
| Strollo [37] | 2020 | Italy | 76.7 ±11.8 | An observational study | ND | ND | 193 | ND | T2D with COVID-19 |

ND: Not Determined;

**Table 2. Risk of bias assessment of the included studies according to the Newcastle-Ottawa Scale (NOS).**

| NOS items / Study ID | Cariou | Fadini | Solerte | Strollo |
|---|:---:|:---:|:---:|:---:|
| Is the case definition adequate? | * | * | * | * |
| Representativeness of the cases | * | * | * | * |
| Selection of controls | * | * | * | * |
| Definition of controls | * | * | * | * |
| Compatibility | * | ** | ** | * |
| Ascertainment of Exposure | * | * | * | * |
| Same method of ascertainment for cases and control | * | * | * | * |
| Non-response Rate | * | * | * | * |
| Total Score | 8 | 9 | 9 | 8 |

## Discussion

Because the effects of DPP-4is on COVID-19 are inconsistent and vague, we conducted this meta-analysis to determine whether DPP-4is could be a treatment for patients with COVID-19 and diabetes. Our meta-analysis may support the hypothesis that the use of DPP-4is may exert a protective effect on COVID-19. Our findings showed that the use of DPP-4is is associated with decreased mortality due to COVID-19 or the risk of progression to mortality (Fig 2).

### Mechanism underlying the relationship between DPP-4is and COVID-19

The mechanisms underlying the effect of DPP-4is on the outcomes of COVID-19 are not clear, but several mechanisms may provide some insights. First, DPP-4is, including sitagliptin, alogliptin, vildagliptin, saxagliptin and linagliptin, are drugs that are widely used to treat diabetes and approved by the Food and Drug Administration [17, 18]. Compared with insulin

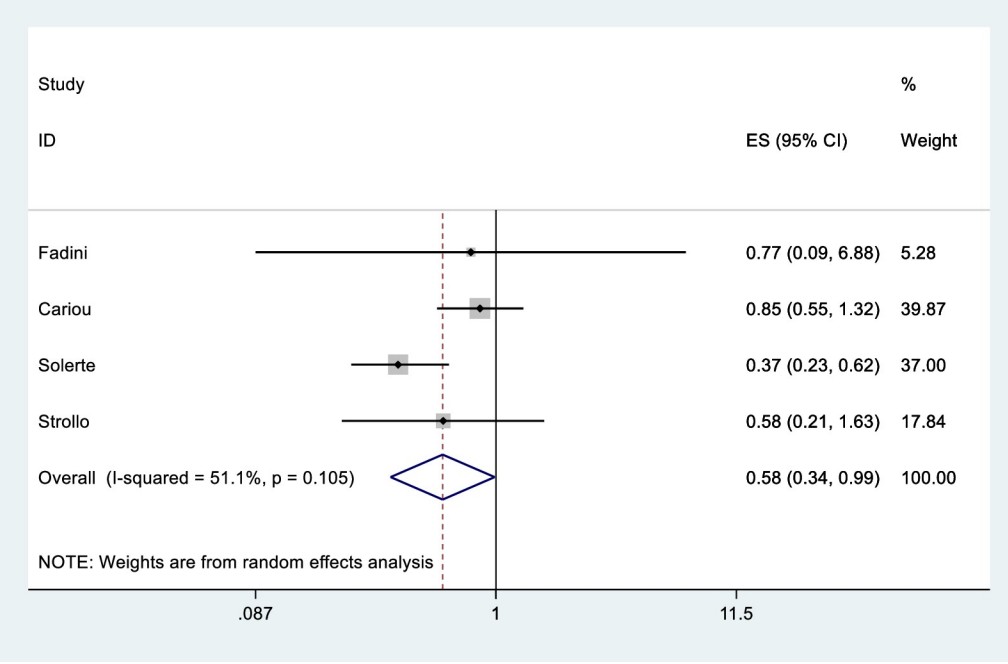

**Fig 2. Forest plots of ORs for the association between the DPP-4i use and the mortality of COVID-19.**

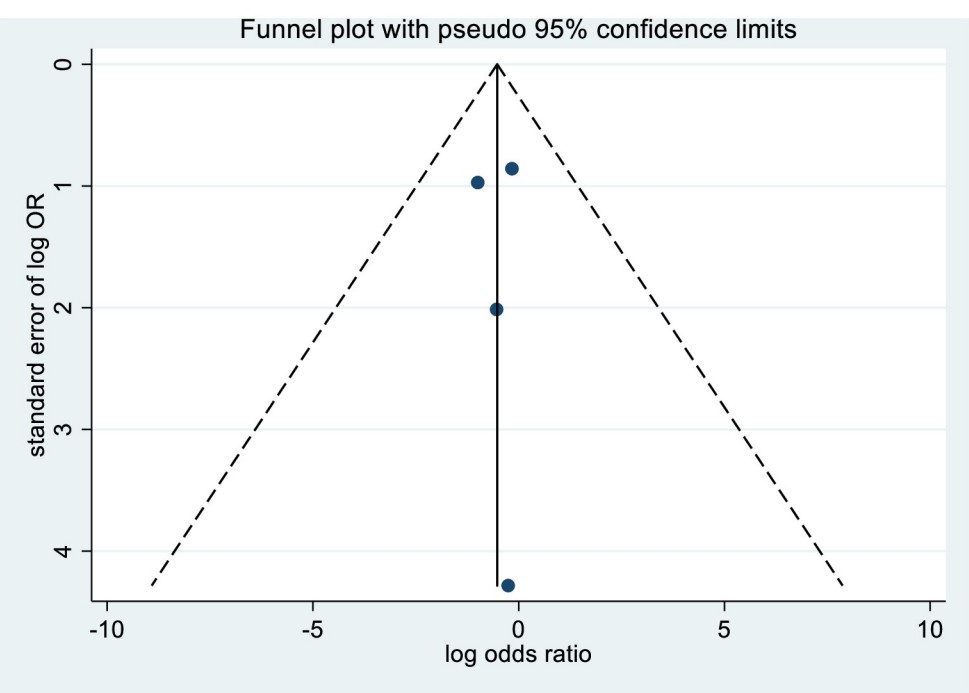

**Fig 3. Funnel plot of the association between the DPP-4i use and the mortality of COVID-19.**

alone, DPP-4is combined with insulin effectively control blood glucose levels, and their effectiveness and safety are guaranteed [19, 20]; additionally, good glucose control can improve the prognosis and outcome of COVID-19 [21]. Therefore, the use of DPP-4is may also exert a beneficial effect on controlling glucose homeostasis in patients with COVID-19 and diabetes.

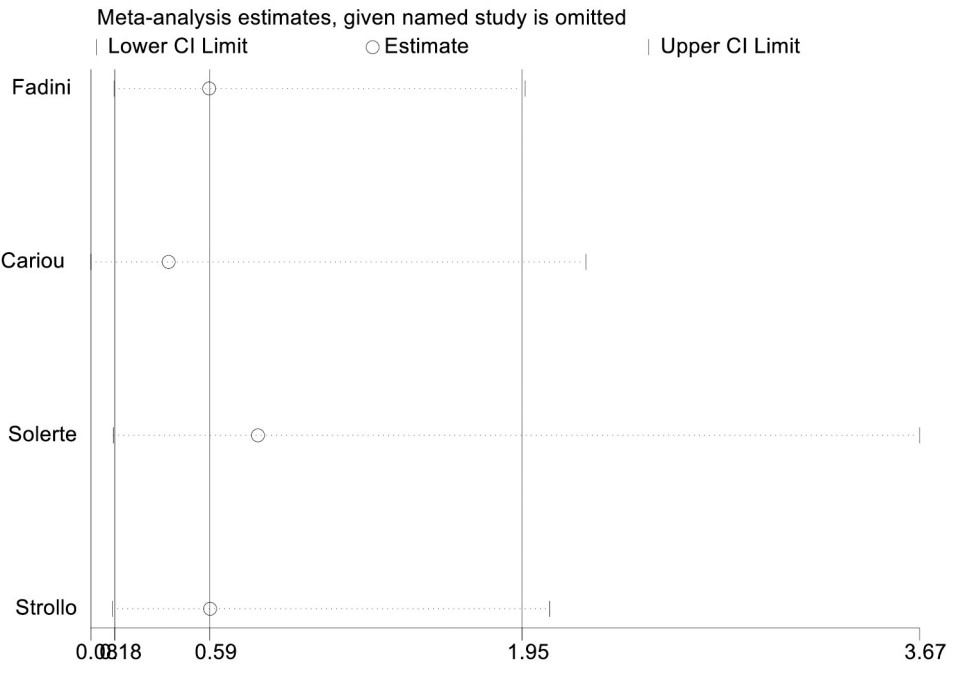

**Fig 4. Sensitivity analysis of the effect DPP-4i use on the mortality of COVID-19.**

Second, DPP-4 has been suggested to be involved in the process of various inflammatory diseases [22, 23]. DPP-4 themselves directly promote T cell proliferation, CD86 expression, activation of the NF-κB signaling pathway and excessive production of inflammatory cytokines to lead to an imbalance of inflammation [9, 24], while severe cases of COVID-19 are characterized by excessive inflammation and the substantial production of pro-inflammatory factors [25, 26]. Moreover, it seems that Sitagliptin has proven efficacy against acute respiratory distress syndrome (ARDS), the common causes of COVID-19-related death, because this drug inhibits IL-6, IL-1, and TNF in individuals with lung injury [27]. In addition, DPP-4is also exert a direct anti-inflammatory effect on the lungs [27, 28]. Therefore, we postulate that DPP-4is themselves play a role in the treatment of COVID-19.

Additionally, glucagon-like peptide 1 (GLP-1), a gut-derived incretin, is secreted after a meal to promote insulin secretion and inhibit glucagon secretion, while GLP-1 not only plays a role in glucose control but also possesses anti-inflammatory properties [29, 30]. Since DPP-4 in the circulation degrades GLP-1 rapidly to maintain glucose homeostasis, the use of DPP-4is may promote the anti-inflammatory effect of GLP-1 and indirectly achieve the purpose of inhibiting inflammation in patients with COVID-19. Although DPP-4 degrades GLP-1 and GLP-1 exerts an anti-inflammatory effect, researchers have not determined whether DPP-4is also inhibit the degradation of GLP-1 in patients with COVID-19. The anti-inflammatory effect is only our conjecture at present, and further proof is needed.

Last but not least, DPP-4 levels are significantly increased in the blood of patients with obesity and obesity-induced metabolic syndrome [31–33], and these comorbidities can aggravate the outcome of patients with COVID-19 [34, 35]. Therefore, studies aiming to determine whether DPP- 4is are useful medicines to treat COVID-19 are important.

## Theoretical and practical significance

DPP-4is are hypoglycemic agents commonly used to treat diabetes, but researchers have not clearly determined whether DPP-4is can continue to be used after a patient contracts COVID-19. For the first time, our study systematically analyzed the effect of DPP-4i use on the mortality of COVID-19 in patients with both T2DM and COVID-19 and found that the use of DPP-4is may exert protective effect on death due to COVID-19. Our research may provide guidance for the treatment of patients with T2DM during the COVID-19 pandemic. Moreover, the relationship between DPP-4 and COVID-19 requires further research.

## Limitations of the study

First, because few randomized controlled studies, case-control studies, and cohort studies on the relationship between DPP-4is and COVID-19 have been performed, the sample size in this meta-analysis was too small and the conclusions are not convincing. Second, as many types of DPP-4is are available, more data are needed to confirm the relationship between the use of DPP-4is and the outcomes of COVID-19; more clinical studies need to be performed.

## Conclusions

In summary, the use of DPP-4is may ameliorate the progression of COVID-19 or the mortality due to COVID-19; this information may help guide the treatment of patients with T2DM and COVID-19, but more well-designed research is urgently needed to support or refute our results.

## Supporting information

**S1 Checklist.**
(DOC)

**S1 File. Full electronic search.**
(DOCX)

## Author Contributions

**Conceptualization:** Yan Yang.

**Data curation:** Yan Yang, Zixin Cai.

**Formal analysis:** Yan Yang, Zixin Cai.

**Funding acquisition:** Jingjing Zhang.

**Investigation:** Zixin Cai.

**Methodology:** Yan Yang, Zixin Cai.

**Project administration:** Jingjing Zhang.

**Resources:** Zixin Cai, Jingjing Zhang.

**Software:** Yan Yang, Zixin Cai.

**Supervision:** Jingjing Zhang.

**Validation:** Zixin Cai, Jingjing Zhang.

**Visualization:** Zixin Cai, Jingjing Zhang.

**Writing – original draft:** Yan Yang.

**Writing – review & editing:** Jingjing Zhang.

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
