## [Decision Letter · Decision Letter 0]

17 Feb 2021

PONE-D-20-34362

DPP-4 inhibitors may improve the mortality of coronavirus disease 2019: A Meta-Analysis

PLOS ONE

Dear Dr. Zhang,

Thank you for submitting your manuscript to PLOS ONE. After careful consideration, we feel that it has merit but does not fully meet PLOS ONE’s publication criteria as it currently stands. Therefore, we invite you to submit a revised version of the manuscript that addresses the points raised during the review process.

We look forward to receiving your revised manuscript.

Kind regards,

Ghulam Md Ashraf, Ph.D., FRSM, MRSB

Academic Editor

PLOS ONE

Journal Requirements:

Reviewers' comments:

Reviewer's Responses to Questions

**Comments to the Author**

1. Is the manuscript technically sound, and do the data support the conclusions?

Reviewer #1: Partly

Reviewer #2: Yes

2. Has the statistical analysis been performed appropriately and rigorously? 

Reviewer #1: Yes

Reviewer #2: Yes

3. Have the authors made all data underlying the findings in their manuscript fully available?

Reviewer #1: Yes

Reviewer #2: Yes

4. Is the manuscript presented in an intelligible fashion and written in standard English?

Reviewer #1: No

Reviewer #2: No

5. Review Comments to the Author

Reviewer #1: DPP-4 Ace receptor are part of the receptor for viral entry. Studies from various European contries especially study from French Cohort has shown that the effect ACE inhibitor is neither harmful nor beneficial. Do they think same type of response from DPP-4 Inhibitor?

Page-3 paragraph 2= DPP-4 are neutral agents but not hypoglycemic agents. Sentence need to be modified.

Page-4, para 2.1= We started article started from...…..= sentence needs to be modified.

Page-6, search results.. what was the probable cause of duplication?

Page-9, paragraph 2 line3= DPP-4 should be replaced from DPP-4 inhibitor

Page-9 Para3= Is this phenomena agent specific or non-specific?

Outcome of results should be mentioned properly not in a sketchy manner.

The stratification of Covid-19 positive patients should have been done by using various tools like modified APACHE-2 and SOFA,MIMS etc.

In Discussion- Inconclusive. Number of subjects very less. It should have been compared with SGLT-2 inhibitors with or without ACE inhibitors.

Reviewer #2: The authors have conducted a meta-analysis of the existing studies on the effect of DPP4i use on COVID-19 mortality in T2DM patients. The findings are of great interest. However, the rationale of the study (SARS-CoV and SARS-CoV2 do not use DPP4 as the receptor, see Cell Study Markus Hoffmann, Hannah Kleine-Weber, Simon Schroeder, Nadine Krüger, Tanja Herrler, Sandra Erichsen, Tobias S. Schiergens, Georg Herrler, Nai-Huei Wu, Andreas Nitsche, Marcel A. Müller, Christian Drosten, Stefan Pöhlmann, SARS-CoV-2 Cell Entry Depends on ACE2 and TMPRSS2 and Is Blocked by a Clinically Proven Protease Inhibitor, Cell, Volume 181, Issue 2, 2020, Pages 271-280.e8,ISSN 0092-8674,

https://doi.org/10.1016/j.cell.2020.02.052.,

Jean K Millet, Javier A Jaimes, Gary R Whittaker, Molecular diversity of coronavirus host cell entry receptors, FEMS Microbiology Reviews, 2020;, fuaa057, https://doi.org/10.1093/femsre/fuaa057),

the presentation of the data (writing of the results section) and therefore the discussion is not adequate

The grammar and syntax are not up to Standard for smooth reading experience. In addition, there are spelling mistakes

6. PLOS authors have the option to publish the peer review history of their article (what does this mean?). If published, this will include your full peer review and any attached files.

Reviewer #1: No

Reviewer #2: No

---

## [Author Response · Author response to Decision Letter 0]

21 Mar 2021

Dear Reviewers,

We were pleased that the reviewers agreed that “the findings are of great interest”. Thank you for the constructive comments and suggestions; we found them helpful for improving our manuscript. We have modified the manuscript according to these suggestions. Our responses to the comments on the manuscript are provided below.

Reviewer #1: DPP-4 Ace receptor are part of the receptor for viral entry. Studies from various European contries especially study from French Cohort has shown that the effect ACE inhibitor is neither harmful nor beneficial. Do they think same type of response from DPP-4 Inhibitor?

Thank you for the constructive comments. We presume that you would like us to include the results of the study by the Carious team [1]. The authors mentioned in their article that a significant difference in the effect of ACE inhibitors on COVID-19 was not observed (OR=1.43 95% CI, 0.99-2.08), and the effects of DPP-4 inhibitors were not significantly different, but they tended to exert a protective effect (OR=0.85 95% CI, 0.55-1.32). Interestingly, when we combined their results with the results of the other included studies [2-4], we ultimately observed a significantly different protective effect (Figure 2).

Page-3 paragraph 2= DPP-4 are neutral agents but not hypoglycemic agents. Sentence need to be modified.

Thank you for the constructive suggestion. We have modified this sentence to “DPP-4 inhibitors (DPP-4is), one of the new oral therapies for diabetes characterized by neutral weight and few adverse effects, is now used to improve insulin secretion as a treatment for T2DM” in the manuscript (lines 8 to 10 on page 3 ).

Page-4, para 2.1= We started article started from...…..= sentence needs to be modified.

Thank you for the kind reminder. We have modified this sentence in the revised manuscript (line 13 to 14 on page 4).

Page-6, search results.. what was the probable cause of duplication?

Thank you for the useful suggestion. When we searched different databases with the same keywords, the same documents or results are often retrieved. Here, we used Endnote to find duplicate results and avoid wasting effort in screening documents.

Page-9, paragraph 2 line3= DPP-4 should be replaced from DPP-4 inhibitor

Thank you for the constructive suggestion. We have replaced DPP-4 with DPP-4 inhibitor in our revised manuscript (line 18 on page 8).

Page-9 Para3= Is this phenomena agent specific or non-specific?

Outcome of results should be mentioned properly not in a sketchy manner.

We thank the reviewer for this constructive suggestion. DPP-4 degrades GLP-1, and GLP-1 exerts an anti-inflammatory effect, but researchers have not determined whether DPP-4is also inhibit the degradation of GLP-1 in patients with COVID-19. The anti-inflammatory effect is only our conjecture at present, and further proof is needed. We have added this explanation to the manuscript (lines 13-16 on page 9).

The stratification of Covid-19 positive patients should have been done by using various tools like modified APACHE-2 and SOFA,MIMS etc.

Thank you for the kind reminder. Unfortunately, since the available literature does not provide sufficient grouping information, we are unable to conduct this experiment based on this constructive suggestion. 

In Discussion- Inconclusive. Number of subjects very less. It should have been compared with SGLT-2 inhibitors with or without ACE inhibitors.

Thank you for this constructive suggestion. We have added the control groups of our included researches in the Table 1. Unfortunately, due to the incomplete raw data of the included articles, it is unclear whether SGLT-2 inhibitors or ACE inhibitors is involved.

Reviewer #2: The authors have conducted a meta-analysis of the existing studies on the effect of DPP4i use on COVID-19 mortality in T2DM patients. The findings are of great interest. However, the rationale of the study (SARS-CoV and SARS-CoV2 do not use DPP4 as the receptor, see Cell Study Markus Hoffmann, Hannah Kleine-Weber, Simon Schroeder, Nadine Krüger, Tanja Herrler, Sandra Erichsen, Tobias S. Schiergens, Georg Herrler, Nai-Huei Wu, Andreas Nitsche, Marcel A. Müller, Christian Drosten, Stefan Pöhlmann, SARS-CoV-2 Cell Entry Depends on ACE2 and TMPRSS2 and Is Blocked by a Clinically Proven Protease Inhibitor, Cell, Volume 181, Issue 2, 2020, Pages 271-280.e8,ISSN 0092-8674,

https://doi.org/10.1016/j.cell.2020.02.052.,

Jean K Millet, Javier A Jaimes, Gary R Whittaker, Molecular diversity of coronavirus host cell entry receptors, FEMS Microbiology Reviews, 2020;, fuaa057, https://doi.org/10.1093/femsre/fuaa057),

We sincerely appreciate this constructive suggestion. Taking into account the latest developments on the role of DPP-4is in the entry of COVID-19 into cells, as you suggested, we have corrected and updated our manuscript (lines 3-4 on page 2 and lines 6-13 on page 3).

the presentation of the data (writing of the results section) and therefore the discussion is not adequate

Thank you for this constructive suggestion. We have modified the descriptions of the presentation of the data in the discussion section (line 5 on page 8).

The grammar and syntax are not up to Standard for smooth reading experience. In addition, there are spelling mistakes

Thank you for this valuable advice; we have modified the whole manuscript carefully. Moreover, we have submitted our manuscript to American Journal Experts for language editing, and the entire manuscript has been revised.

References

1. Cariou B, Hadjadj S, Wargny M, Pichelin M, Al-Salameh A, Allix I, et al. Phenotypic characteristics and prognosis of inpatients with COVID-19 and diabetes: the CORONADO study. Diabetologia 2020;63(8):1500-1515.doi:10.1007/s00125-020-05180-x

2. Fadini GP, Morieri ML, Longato E, Bonora BM, Pinelli S, Selmin E, et al. Exposure to dipeptidyl-peptidase-4 inhibitors and COVID-19 among people with type 2 diabetes: A case-control study. Diabetes, obesity & metabolism 2020;22(10):1946-1950.doi:10.1111/dom.14097

3. Solerte SB, D'Addio F, Trevisan R, Lovati E, Rossi A, Pastore I, et al. Sitagliptin Treatment at the Time of Hospitalization Was Associated With Reduced Mortality in Patients With Type 2 Diabetes and COVID-19: A Multicenter, Case-Control, Retrospective, Observational Study. Diabetes care 2020;43(12):2999-3006.doi:10.2337/dc20-1521

4. Strollo R, Maddaloni E, Dauriz M, Pedone C, Buzzetti R, Pozzilli P. Use of DPP4 inhibitors in Italy does not correlate with diabetes prevalence among COVID-19 deaths. Diabetes research and clinical practice 2021;171:108444.doi:10.1016/j.diabres.2020.108444

---

## [Decision Letter · Decision Letter 1]

15 Apr 2021

PONE-D-20-34362R1

DPP-4 inhibitors may improve the mortality of coronavirus disease 2019: A Meta-Analysis

PLOS ONE

Dear Dr. Zhang,

Thank you for submitting your manuscript to PLOS ONE. After careful consideration, we feel that it has merit but does not fully meet PLOS ONE’s publication criteria as it currently stands. Therefore, we invite you to submit a revised version of the manuscript that addresses the points raised during the review process.

The authors are advised to address the minor concerns raised by one of the reviewers.

We look forward to receiving your revised manuscript.

Kind regards,

Ghulam Md Ashraf, Ph.D.

Academic Editor

PLOS ONE

Journal Requirements:

Reviewers' comments:

Reviewer's Responses to Questions

**Comments to the Author**

1. If the authors have adequately addressed your comments raised in a previous round of review and you feel that this manuscript is now acceptable for publication, you may indicate that here to bypass the “Comments to the Author” section, enter your conflict of interest statement in the “Confidential to Editor” section, and submit your "Accept" recommendation.

Reviewer #1: All comments have been addressed

Reviewer #2: (No Response)

2. Is the manuscript technically sound, and do the data support the conclusions?

Reviewer #1: Yes

Reviewer #2: Yes

3. Has the statistical analysis been performed appropriately and rigorously? 

Reviewer #1: Yes

Reviewer #2: Yes

4. Have the authors made all data underlying the findings in their manuscript fully available?

Reviewer #1: Yes

Reviewer #2: Yes

5. Is the manuscript presented in an intelligible fashion and written in standard English?

Reviewer #1: Yes

Reviewer #2: Yes

6. Review Comments to the Author

Reviewer #1: No Changes required. Manuscript accepted after revision. The revised paper has incorporated all the required changes. After correction the paper has achieved excellence. Hence accepted.

Reviewer #2: The authors have addressed some of my comments.

However, they need to address additional concerns that have arisen due to oversight and/or not adequately addressing them on the revision.

1. Last para on Pg8 of the manuscript, did the authors mean DPP4 or DPP4 inhibitors

2. Reference 27 is not directly related to ARDS although it could be extrapolated from an LPS-induced mouse model. The way it is written, it seems that Sitagliptin has proven efficacy against ARDS

7. PLOS authors have the option to publish the peer review history of their article (what does this mean?). If published, this will include your full peer review and any attached files.

Reviewer #1: **Yes: **Pinaki Dutta

Reviewer #2: No

---

## [Author Response · Author response to Decision Letter 1]

21 Apr 2021

Dear Reviewers,

We were pleased that the reviewers agreed that “the findings are of great interest”. Thank you for the constructive comments and suggestions; we found them helpful for improving our manuscript. We have modified the manuscript according to these suggestions. Our responses to the comments on the manuscript are provided below.

Reviewer #2: The authors have addressed some of my comments.

However, they need to address additional concerns that have arisen due to oversight and/or not adequately addressing them on the revision.

1. Last para on Pg8 of the manuscript, did the authors mean DPP4 or DPP4 inhibitors

Thank you for the constructive comments. We have replaced the “DPP4 inhibitors” with “DPP4” in the revised manuscript (lines 19 on page 8).

2. Reference 27 is not directly related to ARDS although it could be extrapolated from an LPS-induced mouse model. The way it is written, it seems that Sitagliptin has proven efficacy against ARDS

Thank you for the useful suggestion. We have redescribed this sentence as you suggested in our revised manuscript (lines 2 to 3 on page 9).

---

## [Decision Letter · Decision Letter 2]

6 May 2021

DPP-4 inhibitors may improve the mortality of coronavirus disease 2019: A Meta-Analysis

PONE-D-20-34362R2

Dear Dr. Zhang,

We’re pleased to inform you that your manuscript has been judged scientifically suitable for publication and will be formally accepted for publication once it meets all outstanding technical requirements.

Kind regards,

Ghulam Md Ashraf, Ph.D.

Academic Editor

PLOS ONE

Additional Editor Comments (optional):

Reviewers' comments:

Reviewer's Responses to Questions

**Comments to the Author**

1. If the authors have adequately addressed your comments raised in a previous round of review and you feel that this manuscript is now acceptable for publication, you may indicate that here to bypass the “Comments to the Author” section, enter your conflict of interest statement in the “Confidential to Editor” section, and submit your "Accept" recommendation.

Reviewer #2: (No Response)

2. Is the manuscript technically sound, and do the data support the conclusions?

Reviewer #2: Yes

3. Has the statistical analysis been performed appropriately and rigorously? 

Reviewer #2: Yes

4. Have the authors made all data underlying the findings in their manuscript fully available?

Reviewer #2: Yes

5. Is the manuscript presented in an intelligible fashion and written in standard English?

Reviewer #2: Yes

6. Review Comments to the Author

Reviewer #2: My comment

2. Reference 27 is not directly related to ARDS although it could be extrapolated

from an LPS-induced mouse model. The way it is written, it seems that Sitagliptin

has proven efficacy against ARDS

Author's response

Thank you for the useful suggestion. We have redescribed this sentence as you

suggested in our revised manuscript (lines 2 to 3 on page 9).

What the authors wrote

Moreover, it seems that Sitagliptin "has proven efficacy" against acute respiratory distress syndrome (ARDS), the common

causes of COVID-19-related death, because this drug inhibits IL-6, IL-1, and TNF in

individuals with lung injury [27].

My suggestion was that Reference 27 pertains to work with LPS in "mice" which is NOT ARDS in "humans" but may mimic some of the features. Therefore, the authors "CANNOT" make statements such as above. Please correct the statement to accurately reflect the utility of Sitagliptin

7. PLOS authors have the option to publish the peer review history of their article (what does this mean?). If published, this will include your full peer review and any attached files.

Reviewer #2: No

---

## [Editor Report · Acceptance letter]

11 May 2021

PONE-D-20-34362R2 

DPP-4 inhibitors may improve the mortality of coronavirus disease 2019: A Meta-Analysis 

Dear Dr. Zhang:

I'm pleased to inform you that your manuscript has been deemed suitable for publication in PLOS ONE. Congratulations! Your manuscript is now with our production department. 

Kind regards, 

on behalf of

Dr. Ghulam Md Ashraf 

Academic Editor

PLOS ONE